# Complexity reduction by symmetry: Uncovering the minimal regulatory network for logical computation in bacteria

Luis A. Álvarez-García[1]*, Wolfram Liebermeister[2], Ian Leifer[1], Hernán A. Makse[1]*

**1** Levich Institute and Physics Department, City College of New York, New York, New York 10031, United States of America, **2** Université Paris-Saclay, INRAE, MaIAGE, 78350 Jouy-en-Josas, France

\* luisalvarez.10.96@gmail.com (LAÁ-G); hmakse@ccny.cuny.edu (HAM)

## Abstract

Symmetry principles play an important role in geometry, and physics, allowing for the reduction of complicated systems to simpler, more comprehensible models that preserve the system's features of interest. Biological systems are often highly complex and may consist of a large number of interacting parts. Using symmetry fibrations, the relevant symmetries for biological "message-passing" networks, we introduce a scheme, called Complexity Reduction by Symmetry or CoReSym, to reduce the gene regulatory networks of *Escherichia coli* and *Bacillus subtilis* bacteria to core networks in a way that preserves the dynamics and uncovers the computational capabilities of the network. Gene nodes in the original network that share isomorphic input trees are collapsed by the fibration into equivalence classes called fibers, whereby nodes that receive signals with the same "history" belong to one fiber and synchronize. Then we reduce the networks to its minimal computational core via k-core decomposition. This computational core consists of a few strongly connected components or "signal vortices," in which signals can cycle through. While between them, these "signal vortices" transmit signals in a feedforward manner. These connected components perform signal processing and decision making in the bacterial cell by employing a series of genetic toggle-switch circuits that store memory, plus oscillator circuits. These circuits act as the central computation device of the network, whose output signals then spread to the rest of the network. Our reduction method opens the door to narrow the vast complexity of biological systems to their minimal parts in a systematic way by using fundamental theoretical principles of symmetry.

**Citation:** Álvarez-García LA, Liebermeister W, Leifer I, Makse HA (2025) Complexity reduction by symmetry: Uncovering the minimal regulatory network for logical computation in bacteria. PLoS Comput Biol 21(4): e1013005. https://doi.org/10.1371/journal.pcbi.1013005

**Data availability statement:** Network of *E.coli* obtained from RegulonDB; Network of B. subtilis obtained from SubtiWiki. The code and data are available at: https://github.com/luisalvarez96/MinimalTRN; https://github.com/makselab/MinimalTRNCodes; https://osf.io/eh6ps/.

**Funding:** Funding was provided by NIBIB and NIMH through the NIH BRAIN Initiative Grant # R01 EB028157 to H.M. The funders had no role in study design, data collection and analysis, decision to publish, or preparation of the manuscript.

**Competing interests:** The authors have declared that no competing interests exist.

## Author summary

Biological systems are constituted by complex interactions between a large number of different components, and being able to reduce their complexity in order to understand their behavior is of paramount importance. Here we use symmetry principles, in a manner akin to physics, to reduce the Gene Regulatory Networks (GRN) of *Escherichia coli* and *Bacillus subtilis* bacteria to reveal the computational core structure of these networks responsible for driving their dynamics. This computational core comprises gene logic circuits, such as toggle-switches and oscillatory circuits, which ultimately are in charge of the decision making in the bacterial cell. This Complexity Reduction by Symmetries (CoReSym) method opens the way to understanding biological complexity based on firm theoretical principles.

# 1 Introduction

## 1.1 Complexity reduction through symmetries

One of the main challenges of systems biology is that biological systems are inherently complex, as often reflected in the sheer mass of quantitative parameters and details needed to describe such systems accurately and precisely [1]. The human brain, an evident example, consists of $\sim$ 80 billion neurons with 100 trillion connections between them, each one with a large set of parameters defining the strength of their interactions. In the mouse brain, with three orders of magnitude fewer neurons, advanced techniques need to be employed to understand the collective macroscopic behavior [2]. Even in the neural system of *C. elegans* worms, composed of merely 302 neurons, it is not known how this tiny connectome leads to function, and low-dimensional models are needed and regularly used [3,4].

High-dimensional parameter spaces are ubiquitous in biological systems. Finding low-dimensional effective models to describe the dynamics of these systems is crucial to understand how function and collective behavior emerge from the complex dynamics of the system's constitutive elements. This is where concepts and methods from physics have proven to be of great help [5–8]. In physical systems, one often encounters the challenge of handling high-dimensional experimental data. Fortunately, solid theoretical methods have been developed to address this, one of the most powerful being the use of symmetries.

Fibration symmetries [9,10] are symmetries, or redundancies, of the pathways through which signals, or messages, are transmitted, as will be explained further in Section 2.3. These symmetries are identified by finding sets of nodes with identical input trees, known as fibers. Grouping nodes into fibers is particularly useful because it allows for the reduction of network complexity by collapsing symmetric nodes, without disrupting the network's "information flow." This reduction has been applied for understanding the structure of gene regulatory networks (GRNs) in bacteria, enabling us to simplify these networks and present a more transparent understanding while preserving their dynamics [10,11].

Furthermore, a partition of the network's nodes into its fibers allows for the detailed breakdown of these networks into their constitutive *building block* components, their canonical forms in Fig 2 as presented in Ref [10] with examples taken from *E. coli*'s GRN.

## 1.2 Gene regulatory networks (GRNs)

Here we focus on the regulation of bacterial genes as an example of signal processing in a complex biological system. The model bacterium *E. coli* for example, possesses a genome of

more than 4,000 genes (compiled by RegulonDB's aggregate of results to date [12]), 1843 of which are known to regulate other genes through transcription factor (TF) proteins.

The *Gene Regulatory Network* (GRN) is made up of all the transcriptional regulations between genes. When a gene regulates another, via a TF protein that binds to the regulated gene's binding region, this constitutes a directed edge in the GRN between the two (genes) nodes. However, every individual transcriptional interaction between two genes requires a multitude of molecular parameters for a precise mathematical description of the gene expression dynamics [1]. These parameters include everything from transcription and translation rates to binding and unbinding of the TFs as well as ribosomes, for example. In order to accurately describe the global dynamics of the GRNs through ordinary differential equations (ODEs), all of these microscopic parameters would need to be known.

Given that most of these parameters are unknown, structural analyses of the GRNs formed by these "lumped" edges have typically been forced to overlook these heterogeneities, treating all edges representing the same set of parameters and modulo up-regulation and down-regulating effects. In effect, this simplifies the system to two main regulatory types, distinguishing only between two main types, namely, repressors and activators. We continue with this approach; given that it has led to considerable insights in the past, we refer to Section A of the Supporting Information S1 Text for more details on these challenges.

In particular, the discovery of network motifs, small and local motifs that have been identified by statistical overrepresentation in the network, compared to randomized networks that preserve the same observed degree distribution [13,14]. Although the individual dynamics of some network motifs is more or less understood [13–15], given that they are in essence local structures, they do little to unravel the global topology of the network or the global dynamics.

On a large scale, it has been proposed that the *E. coli*'s GRN has a feedforward structure [13,16–19], where signals flow unidirectionally from a core of sensors and master regulators through a series of parallel layers down to an outer periphery in a feedforward manner [13,14,19]. The modular structure of the network has also been noted [19]. However, there also exist a significant number of feedback loops, which complicate this picture. Thus, many questions remain unanswered. What is the relevant structure at the "center" of these systems. What is the core structure responsible for decision making? Which genes belong to this computational core? How does this structure control the rest of the network? Is there a minimal computation core that explains the structure and function of the GRN in a simplified manner?

### 1.3 Gene regulatory networks as a computing device

We can interpret a transcriptional regulation from one gene to another as a form of "transcriptional signal" that one gene sends to another. Representing a regulatory "message" with the TF as a "messenger." The pathways of the GRN can then be seen as "signaling pathways," on aggregate establishing the "signaling flow," or "information" (loosely stated) flow in the network. A main feature of GRNs, which we emphasize here, is that signals do not only propagate in "forward direction" between different layers of the network, but can also cycle in feedback loops. This is significant because network structures that only allow forward transmission, or sequential logic, map input signals to output signals of similar shapes, possibly blurred, inverted, or time-delayed [1]. They may also aggregate several inputs and generate several outputs, but it is not possible for them to have a memory of previous states. Circuits in which signals can cycle show more complex behavior: they can stabilize an output variable, generate oscillations, and may internally store information, like a toggle switch in synthetic biology [20] analogous to a flip-flop in electronics.

In electronics, feedforward circuits are called "combinatoric logic circuits" and are memoryless digital circuits whose output at a given time depends only on the combination of its inputs. These circuits are made of standard logic gates such as NOR and NAND. On the other hand, circuits with feedback are called "sequential logic circuits." Their output depends on both their present input and also their previous output. This feedback loop provides them with memory, since the circuit is able to "remember" its state even when the external input is removed. Combinatorial and sequential circuits are the "decision-making" machinery behind the logic function of electronic circuits.

Even though a gene's expression level is in fact a continuous quantity, a Boolean logic approximation can help picture it's expression levels. Indeed, a GRN modeled under a Boolean approximation can simulate any finite state machine [21] (a simple sequential computing device with memory of its state). Furthermore, given the important role of GRNs in the regulation of the reaction of bacteria to both its internal and external states, it is reasonable to assume that they would play a central role in their *decision-making* dynamics. So, if the cell is to perform as a form of biological computational device (in the sense that it needs to "output" a reaction in response to the external "inputs" it encounters and its internal "states"), then both forms of sequential and combinatoric logic circuits should play an important role in the GRN. Indeed, we find these two modes of signal transmission appear both on a small and on a large scale in the GRN.

## 1.4 Complexity reduction through fibrations: constructing a minimal TRN

In this work, we describe how to reduce the complexity of GRNs to their "minimal computational core" by applying the novel tool of fibration symmetries, in addition to standard graph-theoretical tools. Based on this notion of symmetry, we present a novel method to reduce any directed "message-passing" network (any network where edges represent signals between nodes) to what we call its computational core. After the GRN is reduced to its minimal structure, we analyze this minimal structure. The overall procedure consists of five steps.

Steps (I) and (II) are concerned with the reduction of the network, removing all its elements that do not contribute to its computational capabilities. Reducing the GRN to the core network at the heart of the decision-making processes: the minimal GRN. Step (I) eliminates all redundant information pathways through the use of graph fibrations. While step (II) removes the nodes that only receive signals or just pass-it-through without contributing to the decision-making process via the $k$-core decomposition, these nodes are responsible for communicating the output from the minimal GRN to the peripheries. Steps (III)-(V) analyze this minimal GRN. In step (III) we focus on the large-scale structure of the minimal GRN: how the components of this core network are connected with each other. The last two steps "zoom-in" to look at the small-scale, or local, structures within the minimal GRN's components, by looking at the logic circuits (step IV) and how these are connected with each other as well as informing about the connectivity structure within the different components (step V).

Steps (I) and (IV) decompose the genetic network into its building blocks by using fibration symmetries (Fig 2) and broken symmetries, respectively. The fiber building blocks correspond to the 3 basic canonical types shown in Fig 2: I) simple $n$-ary trees $|n, \ell\rangle$, II) Fibonacci building blocks and III) Composite Building Blocks.

## 1.5 The structure of the minimal GRN: large-scale components and small-scale, local circuits

We apply our method to model bacteria *E. coli* and *B. subtilis*. In the case of *E. coli*, reducing it from the entire GRN to the left in Fig 1 to the much simpler network to the right (representing a sketch of its computational core). The resulting minimal GRNs correspond to just a tiny fraction of the total genes and are composed of vortices on a large scale and feedforward and feedback circuits on a small scale. Recognizing this structure helps us to understand what the network can do and what the functions are of different parts of the network. Hence, we not only reduce the GRN to its core computational structure (by omitting symmetric and peripheral nodes), but also identify the "signal vortices" and the local gene circuits that process signals and store information and are, therefore, capable of computations, comparable to a silicon-based computer.

On a large-scale level, the core network of any directed network consists of Strongly Connected Components (SCCs) connected between them in a "forward signaling" manner, along with the nodes that regulate them. See, for example, the right panel of Fig 1, which represents the core network at the heart of *E. coli*'s GRN: the minimal GRN. The SCCs correspond to "signal vortices" in which signals can cycle and then gets sent to the fibers outwardly of this minimal GRN to the periphery. Embedded within these SCCs are the small-scale structures, the logic circuits of the GRN: symmetry-broken memory-storing

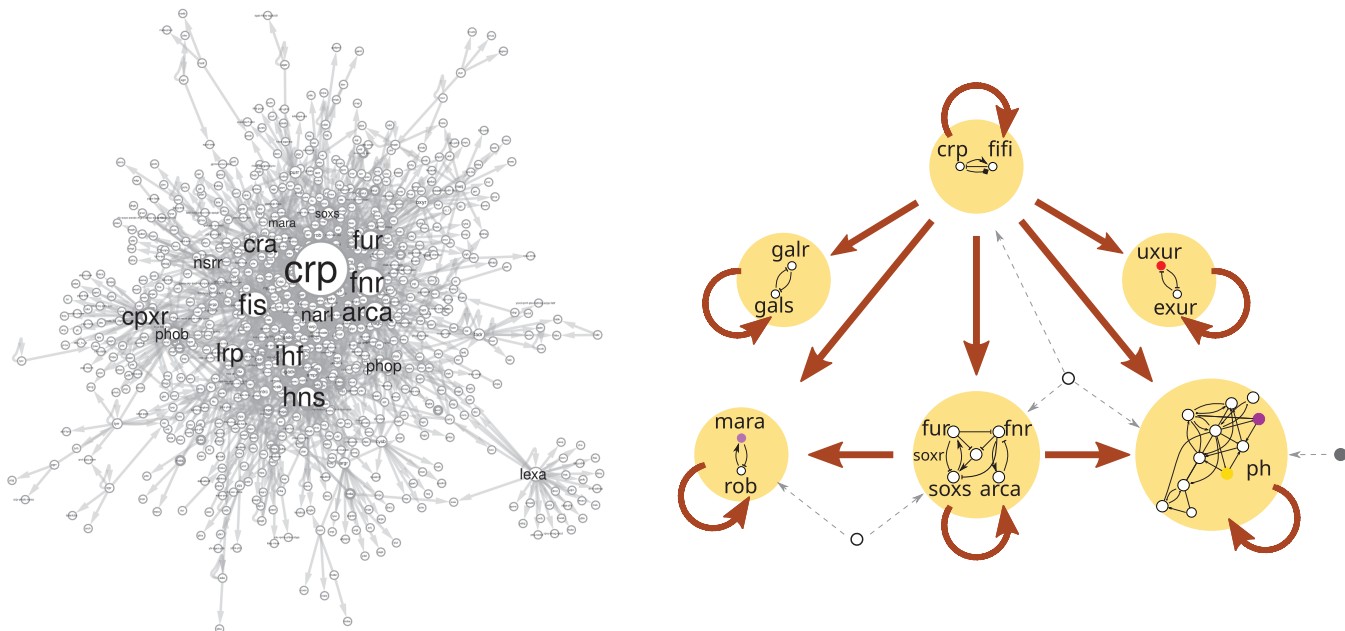

**Fig 1. A minimal GRN obtained through the CoReSym reduction method**. Reduction of Gene Regulatory Network (GRN) of *E. coli* bacteria. Left: 879 nodes operon-GRN of *E. coli*. Here we show only the weakly connected component, i.e.small disconnected pieces of the network are not shown, since they do not play a significant role for the network dynamics. Node sizes and font size are proportional to out-degree of the node. Right: a representative of the minimal GRN obtained after the application of our CoReSym reduction method. The simple depiction of the core network illustrates the signal flow between its different components (bigger "nodes"), the strongly connected components of the network (SCC: a component of a graph that within it, each node can be reached from every other node). The smaller gene nodes inside these SCCs form the computational core of the network. Colored nodes represent collapsed-fiber nodes. Bigger arrows represent the edges between the components. The three nodes outside the components are representative of controller nodes, which send signals to different SCCs. Interestingly, two parallel feedforward structures exist between the components: the central *crp-fis* SCC regulates the *soxS* SCC. In one feedforward structure, they regulate jointly the pH SCC and in the other one the *mara-rob* SCC.

toggle-switches and oscillators; the two primary components of any computer [22]. The system of SCCs represents the smallest computational subunits that cannot be further reduced by fibration symmetries or by smaller strongly connected components. Thus, this structure represents the "minimal GRN" structure of the cell. Thus, the GRN can be seen as a computational machine, where the memory is stored and controlled by broken symmetry circuits within the SCCs.

## 2 Methods

### 2.1 Symmetries and fibrations

Symmetries play a crucial role in physics by simplifying complex models. Noether's theorem established a profound connection between the symmetries of a Lagrangian and its conserved physical quantities in classical mechanics, such as energy and momentum. By identifying and exploiting these symmetries, physicists can reduce the number of variables needed to describe a system while preserving its essential characteristics. This approach has become ubiquitous in physics when dealing with complex problems, from classical mechanics to quantum mechanics and particle physics, where symmetries underpin the Standard Model under the gauge symmetry group U(1)×SU(2)×SU(3), for example.

Motivated by the success of applying symmetries and geometry across various fields, we ask whether a similar approach can be used to tame biological complexity and, if so, in what specific way. For example, what are the symmetries of a system such as *E. coli's* GRN, left panel in Fig 1? As we will explore, symmetries can help us understand these systems, but with an important distinction: the symmetry groups used in physics cannot be directly applied to biological systems. Instead, a new type of symmetry, based on *graph fibrations* [9,23], turns out to be more suitable.

Since we are dealing with networks, the initial assumption would be to look at automorphisms, transformations of a graph that leave it unchanged. However, automorphisms are too restrictive for understanding biological signaling networks [4,10,11,24]. The symmetries that have been so fundamental in physics do not translate well to the complexity of biochemical systems, necessitating a different approach.

Why, then, do automorphisms fail in biology? One could argue that a fully symmetric graph, while significant in physics, maybe less useful to represent a biological system on account of its sensitivity to small alterations which are commonplace in biology. A biological signaling network captures the flow of information dictated by the dynamic interactions between biological units—whether neurons in the brain, genes, or enzymes in a genetic or metabolic network. The information that reaches each node is determined by its inputs alone, not its outputs. Consequently, the full structure (including both input and output) is not crucial for understanding the function of a node in these networks.

In our particular case of gene networks, the flow of signals is well defined. Because the dynamics of each individual component is only determined by its inputs, this begs the question: What if instead we focus only on the inputs of a node? The "history" of the signals, transmitted through the pathways of the network, or input trees, and their symmetries.

As we will explain, nodes with identical input trees will, in fact, share the same dynamical state, a form of symmetry. In other words, nodes that share identical input "history" are symmetric in a certain sense and become synchronized. This is significant because cluster synchronization is ubiquitous across biological systems, from gene expression [25,26] to brain activity [2,5]. The overall function of cells must be driven in part by coherent communication between their units, and this synchronization is captured by the fibration symmetries of the underlying biological graph, as we will show. Thus, the existence of symmetries in the

information pathways of the network helps explain and predict the function of the network, linking its structure to its behavior.

## 2.2 Gene Regulatory Networks (GRNs)

Gene expression in cells is regulated in response to the cells' environment and internal state. In bacteria, gene expression is governed by global mechanisms (e.g. via sigma factors, varying activity of the global transcription and translation machinery), where also "regional effects," such as the chromosome correlated expression of nearby genes on the chromosome, play a role. However, an important part of gene regulation in bacteria occurs through transcription factors (TFs), proteins that bind to DNA sites called promoters that regulate specific genes.

Transcription factors can activate or repress the transcription of genes, thus increasing or decreasing the expression levels of their target genes. The transcription factors activities depend on their own expression levels and can be modulated by small-molecule binding. By binding to different TFs, metabolites can modify their activities and thus modulate the expression levels of their target genes. The metabolites may come from the outside environment of the cell or may be products of the metabolism of the cell itself, providing a feedback mechanism in which the metabolic state of the cell informs the regulation of transcription [1,18,27].

The Gene Regulatory Network (GRN), here understood in a narrow sense, represents the regulatory effects between Transcription Factors (TFs) by regulation of transcription. In the network, if a TF (encoded by gene A) can bind to the promoter region of a gene B and regulate its expression, this is represented by a directed link from gene A to gene B. The type of edge corresponds to the type of regulation being performed by the transcription factor, being activation, inhibition, or dual (depending on how the TF binds to the promoter region). Together, all such regulations form the GRN, which determines the expression of individual genes according to the cell's sensed environment and its own internal, for instance metabolic, states.

We can see here how the gene regulatory system needs to "read" inputs from the environment so that it can then "respond" in an appropriate manner, response that needs to be "signaled" to the rest of bacteria's components in order to enact the appropriate reaction. This system helps us elucidate the needs that bacteria have for "decision making," which will be an important idea in this work. This process of sending regulatory messages as signals across genes defines the information flow in the gene network. The signal can be though of as an "information package" or "message passing" being sent from the source gene to the target gene. See Section A of S1 Text for a more thorough explanation.

## 2.3 Graph fibration formalism

The network reduction method (Steps I and II) explored in this paper is based on graph fibrations. Fibrations were introduced by Grothendieck [28] in the context of category theory and algebraic geometry. Although the original work applies to fibrations between categories and it remains a bit obscure for pedestrians, fortunately, this work has been adapted to graph fibrations by Boldi and Vigna [9] in computer science. Their inspiration was a distributed system of computer processors that need to be synchronized in clusters in a coherent manner for proper global updates, as there is no point to have a processor waiting for its update while being out of sync with the rest. A computer system seen as a graph of processors with fibration symmetries then guarantees coherent optimal processing. As stated in the illuminating words in Vigna's blog on fibrations https://vigna.di.unimi.it/fibrations:

*"If a graph G is used to represent the structure of a network that exchanges messages, and the processors of the network execute the same algorithm starting from the same initial state, the existence of a fibration φ : G → H implies that, whatever algorithm is used, there are executions in which the behavior of the nodes in G is fibrewise constant (i.e., all processors in the same fiber are always in the same state)."*

The concept of fibrations can tell us about the signal processing dynamics and synchronization in networks based on network structure alone [23]. As such, it is crucial to understand cluster synchronization [29,30] (gene coexpression in our case) and the signal processing tasks performed by these networks.

Under some ideal assumptions, this idea can be directly translated into GRNs to help us understand how it might function as a computing device. This was enough inspiration for us to look for fibrations in these systems in the first place [10].

Concretely, the condition that every processor executes the same algorithm is translated to the GRN as the condition that every input function has the same parameters (every edge represents the same equation, module repressor/activator type). This condition is natural for computer processors, but controversial for biology, as discussed above and in more detail in Section A of the S1 Text. Still, as usually done by physicists (inspired by the metaphor of the spherical cow, the legend of the Gordian knot, and Occam's razor), we translate this simplification to biology and try to understand the consequences later. Alas, the cow is not spherical, but in the absence of the "perfect" approach (or any approach at all), it is better to start with a sphere and then introduce details and refinement, as needed. Otherwise, we might risk losing the forest for being too concerned with how the leaves of the tree look.

Another assumptions is that different pathways do not experience significant communication delays which would cause asyncronicity. Additionally, we have a Boolean logic approximation in mind throughout. Its important to note that, unless the input functions (and microscopic parameters therein) drastically change from gene to gene, when these assumptions are not met, our approach is not broken but instead of a "fiberwise constant" behaviour (i.e. gene clusters turning "on" or "off" in unison), we would expect to observe gene coexpression levels. In fact, we have found that fibration symmetries are actually able to predict gene coexpression (or correlated) patterns [11].

## 2.4 Graph fibrations, input trees and fibers

A *graph fibration* is a graph morphism $\varphi : G \rightarrow B$ between a graph $G$ (the total space) and a graph $B$ (the base), in which, for every (pre-image) node $i$ in $G$, and every (image) edge $e'$ in $B$ targeting the image of $i$ (i.e. $i' = \varphi(i) = t(e')$), there is a *unique* edge $e$ (in $G$) targeting $i$ ($i = t(e)$) whose image is $e' = \varphi(e)$. Here, $t(e)$ denotes the *target* node of edge $e$, for more mathematically rigurous definitions we refer to Section B of the S1 Text.

Simply put, every edge targeting an image node $i' = \varphi(i)$ can be *uniquely* lifted to an edge in $G$ targeting its pre-image $i$. This condition is called the lifting property [9]. Crucially, this means that the inputs of any node are preserved in the base graph. In Fig 3A we illustrate the definition of graph fibration, we see examples of three different morphisms, one is not a fibration while two of them are: a surjective fibration, and an injective fibration.

The question now then becomes how to indentify the synchronous clusters from the network structure. To do this we have to look at the input history of each node via their *input tree*. Nodes with identical input histories will become synchronous since they receive the exact same history of signals.

The *input tree* $T_i$, for a node $i$ in graph $G$, is made up of the set of all pathways in $G$ ending at node $i$ [10]. In order to construct the input tree of a node, we start by constructing its

*input set:* the set of incoming edges to the node in question, along with their respective source nodes (see Fig 3B). We then attach to this rooted tree, the input sets of the incoming nodes, and so-on recursively, to obtain the input tree. Therefore, the input tree of a given node summarizes the regulatory pathways of the network that reach this node. This allows us to group and classify the nodes based on the '*history*' of the signals they receive. Fig 3C shows the input trees for the nodes in graph *G*.

Upon determining the sets of nodes with *isomorphic input trees*, we determine the *fibers*: sets of nodes with *identical* input history. Fibers are called *balance colorings* or *equivalent relations* in other branches of mathematics, dynamical systems and chaos [31,32]. For infinite input trees it suffices to show the isomorphism up to $N_G - 1$ layers of the trees (where $N_G$ is the number of nodes in *G*) to determine the isomorphism [33]. The nodes in a fiber receive the exact same signaling tree, which makes the paths of the signals rooted at them redundant, revealing a sense of symmetry. This in turn makes the nodes in the same fiber symmetric in terms of signal processing or information flows in the network.

Having the set of fibers, we can construct the fiber building blocks of the network, Fig 2 shows their canonical forms. These are constructed by obtaining the induced graphs of each fiber along with their regulators (also including the nodes along the path(s) forming a feedback loop from the fiber to the regulators, if any).

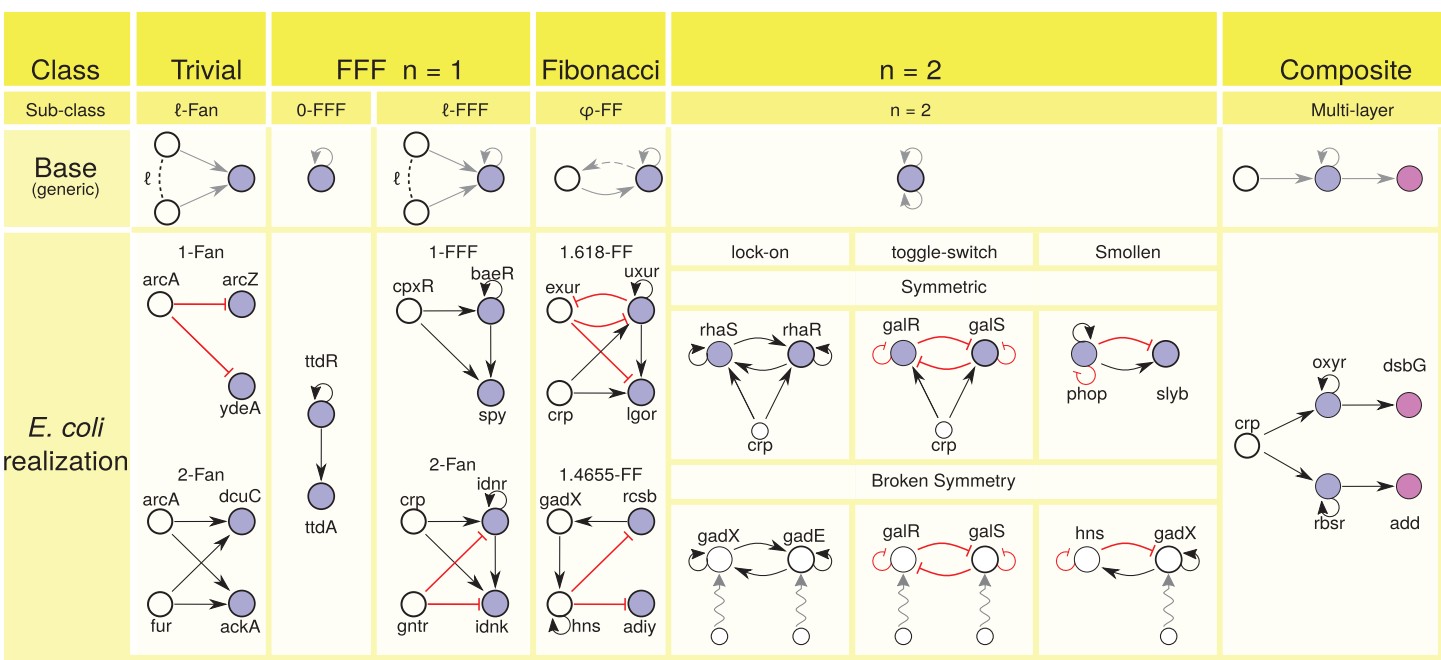

**Fig 2. Canonical fiber building blocks.** These correspond to the canincal fiber building blocks observed in the GRNs of *E. coli* and *B. subtilis*, with examples taken from *E. coli*. The networks can be seen as assemblies of 5 basic classes of fibration building blocks: (i) Trivial fibers. A number $\ell$ of external regulators identically regulate the genes in a fiber, which then show synchronous dynamics. Operons with only one promoter belong to this class, where colored nodes represent genes belonging to the operon (perhaps with more colored nodes in the fibers, depending on the number of genes in the operon). (ii) The feedforward fiber and its sub-classes of $\ell$-FFF with $\ell$ external regulators. The FF fiber is defined by a feedforward motif with a self-loop in the synchronous set of genes, and the number of $\ell$ external regulators. (iii) The Fibonacci fiber, $\varphi$-FF. A more complex building block, defined by a fractal dimension branching ratio that occurs given the presence of a self-loop and a feedback regulation from the fiber back to the regulator(s). The Fibonacci fibers observed in *E. coli* have a branching ratio between 1 and 2, placing this building block in between the FFF fibers and the n=2 fibers. (iv) The n=2 fibers, defined by two self-loops in the synchronized genes. When this symmetry is broken it forms the memory and oscillatory logic circuits embedded in the SCCs. And finally (v) composite fibers of the previous ones. By adding different types of the previous 4 building blocks, in a sequential manner, a composite fiber is obtained. An interesting consequence of this is the synchronization of genes that may be far apart from each other and don't share any regulation.

The notion of fibrations can be extended to graphs with various types of edge, such as a GRN, which can have edges corresponding to several types of interaction: activation, inhibition, other types of interaction. In this case, for two input trees to be isomorphic not only do they need to have the same topology, but the type of edge must be the same.

In the case of gene regulation networks, fibers are sets of genes that are coexpressed in their activity; this is the case for the blue nodes in Fig 3A.

## 2.5 Complexity Reduction by Symmetries (CoReSym) method step by step

We developed a stepwise method called "Complexity Reduction by Symmetries" (CoReSym) to reduce any signaling network to its computational core. The method aims to clarify the structure of the network, help to understand the decision-making processes performed by the network. CoReSym can be applied to any directed network, even outside biology, in which edges represent a signal transmitted from one node to another and provides insights about the collective dynamics within the network based solely on its topology. Regardless of the exact model, it can be used to describe the underlying dynamical system since the method does not depend on the actual form of the admissible ODEs in the graph [9,23,31,32].

The CoReSym method for network reduction consists of five steps. A more detailed description is given in Section C of the S1 Text.

**2.5.1 Steps I and II: Reducing a signaling network to its computational core.** Step I, *'collapsing'*, removes the symmetries in signaling flow in the network that originate from fibration symmetries. This step is based on Lemma 5.1.1 from Ref. [23] which proves that the dynamics of a network is preserved when all symmetries are eliminated by a surjective graph fibration.

In a graph fibration, multiple edges targeting the same node in *G*, cannot be collapsed to fewer edges in *B*, nor can new edges targeting the image of a node be added. Crucially, though, nodes *can* be collapsed if they belong to the same fiber (given their redundant, or symmetric, input trees). This is the crucial feature of graph fibrations, once we have identified the redundant pathways, we can eliminate the redundancies without losing any "information pathways"; only the redundancy is removed while the dynamics are preserved. This is done by collapsing the fibers into a single representative node for each fiber. Such a surjective fibration is shown on Fig 3A.

Step II, *'pruning'* the loose ends of the network, makes use of the direction of the signal flow in the network. An injective fibration (also shown in Fig 3A) formalizes the idea that under certain conditions, a subset of the constitutive elements of a system may drive the dynamics of the entire system. Lemma 5.2.1 in Ref. [23] shows that the dynamics of the nodes in the "outer" layer is driven by the dynamics of the (inner) core network. Therefore, the dynamical behavior of the "core" network can be studied separately and further used to scrutinize the dynamics of the "outer" layer.

Reducing the network to its core is performed by applying an inverse injective fibration: the *k*-core decomposition of the network, which identifies an "outer" layer (shell or periphery) of nodes that do not send signals to the "inner" core of the network, Fig 3D shows this decomposition. This step can be thought of as trimming the loose ends of a tree. The removed nodes belong to the "null"-paths, the dead-end paths in the network: the nodes that do not posses any output, along with the nodes that exclusively regulate them, and so on iteratively. These are the nodes that send signals to the peripheries of the network and other parts of the bacteria, such as the metabolic network via genes that express enzymes for metabolic functions, such as sugar consumption, among others.

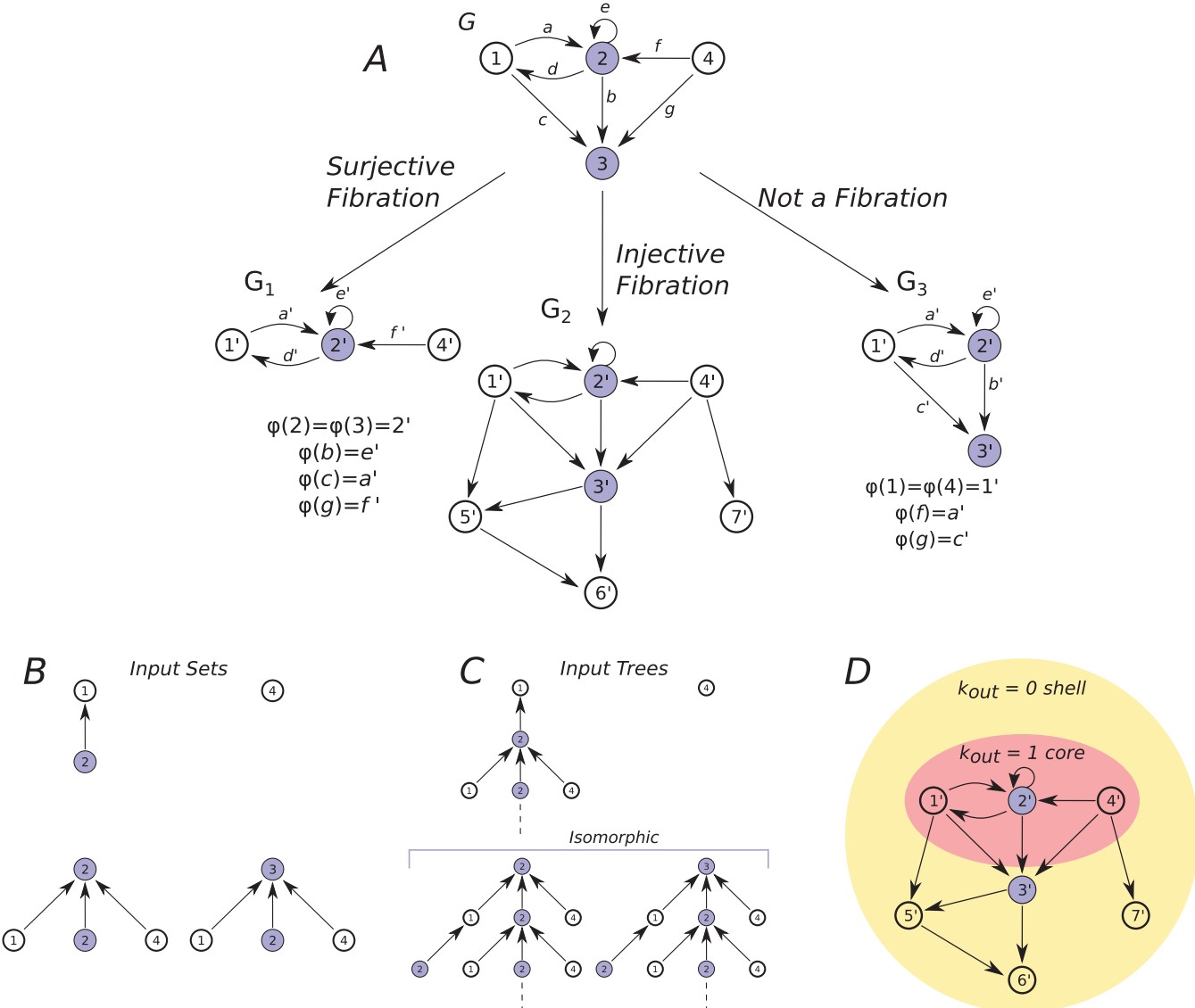

**Fig 3. Fibration and $k_{out}$-core decomposition.** (**A**) Graph $G$, a subgraph of the GRN of *E. coli*, shows a Fibonacci building block with class number $|\varphi = 1.6180, \ell = 2\rangle$ [10,11]. All three mappings are morphisms since the images of all the nodes in $G_1$, $G_2$ and $G_3$ are connected only when corresponding nodes in $G$ are connected, respecting the incidences. The mapping $G \to G_1$, in the left, corresponds to a surjective fibration: all nodes with isomorphic input trees are collapsed to one (nodes 2 and 3 collapsed to $2'$), all input trees are preserved, hence the lifting property is satisfied. Mapping $G \to G_2$ is an injective fibration. Indeed, it is easy to see that the original graph is embedded in $G_2$ making this map a morphism where all input trees are preserved. Some nodes and edges are added but without breaking the original input trees. The mapping $G \to G_3$, which maps node 4 to $1'$ does not correspond to a fibration given that the input-tree of node 4 (seen on **B**) is not preserved in its image node $1'$ in graph $G_3$, the same problem occurs with the images of nodes 2 and 3 ($2'$ and $3'$ respectively), their input trees are not preserved as the former input from node 4 is lost. Edges $a'$ and $c'$ cannot be uniquely lifted at $\varphi(2)$, since they need to be lifted to $a,f$ and $c,g$, respectively, for the mapping to be a morphism. In practical terms, since the input from node 4 is lost, graph $G_3$ represents an entirely different dynamical system from graph $G$. If the graph $G$ represents a GRN, genes $2'$ and $3'$ in $G_3$ would have a different expression pattern than genes 2 and 3. (**B**) Shows the input sets and (**C**) the input trees of nodes in graph $G$. The input set of node 2 is repeatedly attached to node 2 in every layer of the trees, due to its self-loop, this process is repeated ad infinitum. As a result, the input trees of nodes 1, 2 and 3 are infinite; however, since $G$ has only 4 nodes, it suffices to verify the isomorphism up to the third layer of their trees, hence nodes 2 and 3 are determined to have isomorphic input trees. (**E**) Example of the $k$-core decomposition of graph $G_2$ from (**A**). Even though node $5'$ on the outer $k_{out} = 0$ shell (in Yellow) does have one output, once nodes $6'$ and $7'$ in the shell are removed, it will then be left with no output and will be removed as well. All the remaining nodes in the $k_{out} = 1$ core have at least 1 output after doing this process.

**2.5.2 Step III: large-scale structure of the minimal network: strongly connected components and signal vortices.** The previous two reduction steps yield the core subnetwork that controls the dynamics of the entire system, the minimal network. After pruning the loose ends of the network, since all the dead-end paths are lost, all the remaining paths self-cross at some point. The minimal network will therefore always consist of the SCCs and the nodes regulating them (assuming that there are SCCs; for an acyclic network no minimal network would be obtained). For this reason, we want to understand how the network decomposes into SCCs, the large-scale structure of the minimal network. Here, SCCs represent the smallest computational subunits that cannot be further reduced neither by the fibration symmetries nor by smaller strongly connected components.

Hence, on Step III, we decomposed this minimal network into its SCCs. The nodes that do not belong to the SCCs are connectors between them or controllers (external regulators) nodes that send signals to the SCCs but do not receive any signals back from them (otherwise they will be part of the SCC by definition). As a consequence, the most trivial result that could be expected is that the core of any network corresponds to only one single SCC, i.e. virtually no structure in the core network. This is in fact the case for most randomized (degree-preserving) versions of these networks as will be shown later in more detail (Section 3.4), but not at all the case for the GRNs studied.

**2.5.3 Steps IV and V: Small-scale structure inside the SCCs: logic circuits and cycles.** The last two steps consist of understanding the small-scale structures within the SCCs: the logic circuits (Step IV) and the bigger cycles connecting them (Step V).

Since the inception of synthetic biology and the first genetic circuits designed two decades ago [20,34–37], it is known that simple genetic circuits can perform the basic logic operations necessary for any computational device, such as memory storage and timekeeping [34,35,38–40]. Such circuits are constructed using feedback loops (and hence they will always be embedded in the SCCs), both positive and negative [38,40], and are executed by synthetic switches and oscillators designed from simple components such as interacting genes or protein-protein interactions [38,40].

The most basic memory circuit corresponds to the toggle-switch [20], analogous to a bistable flip-flop in electronics [22] that stores one bit of information given that it has two possible stable and reciprocal states. It is comprised by two mutually repressive (MR) genes with different inputs for each gene, the *'set' (S)* and *'reset' (R)* switches. While for timekeeping, oscillating circuits can be obtained by a "frustrated" signaling chain, most commonly by a negative feedback loop (NFBL) driving the system back and forth between the stable steady states [38]. The most simple form of this is two genes with a NFBL between them, where one gene activates the other, while at the same time it is being inhibited by the other. The presence and type of self-regulations in these cases changes the specific dynamics: no self-regulations requires noise to drive the oscillations [41,42], while the most robust version corresponds to the Smollen oscillator [35]. Other forms of oscillating circuits are also possible [24,34,39,40]. These circuits have been known from before, but here we are able to reinterpret them as broken symmetry versions of symmetric fiber building blocks, see Section C in S1 Text for further explanation.

In Step IV we systematically look for circuits capable of logic computations in the minimal network. Since these circuits can be artificially constructed to perform computations, it is reasonable to expect to observe them, or some close variation, at the core computational subset of the network. We would expect to find both memory storage circuits as well as oscillating circuits for timekeeping. In the case of the GRN of simple model bacteria, we would expect to observe the simpler forms of these known genetic circuits from synthetic biology.

Indeed, we find the presence of circuits closely resembling all these circuits in the minimal network driving the GRN. This suggests that we can understand this minimal network as a logical computational machine.

Furthermore, in Step V, we study the structure of the signal vortices, the SCCs, that make up the large-scale structure of the minimal GRN and the interconnectedness between the different logic circuits present. An SCC is composed of a complicated arrangement of feedback loops between its constituent nodes, as such we probe its structure by studying the independent simple cycles present in the minimal GRN. These cycles in themselves represent a form of longer-term memory, responsible for the interconnectedness of the logic circuits, where signals loop between different logic circuits.

Crucial to the dynamics of these circuits are feedback loops between different genes. This implies that the circuits are always embedded in the SCCs of the network. Given that the SCCs are preserved after our reduction method, we know that we are not losing any logic components of the network as we reduce it. This means that we can interpret the SCCs of the network as the modules where the logic computations are performed.

## 3 Results

### 3.1 Application of CoReSym to bacterial gene regulatory networks

To demonstrate the use of our method, we applied it to two of the most widely studied bacterial GRNs, the networks of *E. coli* [13,14,16,18] and *B. subtilis* [43,44]. Their step-wise reduction is shown in Fig 4, while some statistics are given in Tables 1, 2, and 3.

We observe a rich structure of connections between multiple SCCs, both direct connections and through longer pathways crossing *bridge* or *connector* nodes. This suggests that the structures we observed are not the result of randomly generated networks, based on the fact that the number of SCCs for each GRN is more than 5 standard deviations away from the average number of SCCs observerd in the randomized degree-preserved generated networks (more on Section 3.4), indicating it is extremely unlikely for these structures to originate out of pure chance.

In both bacteria studied, we obtained a rich but transparent structure: the SCCs of the network receive inputs from outside controlling nodes as well as some connecting nodes between the different SCCs, responsible for transmitting signals between the SCCs. This also yields a simple and modular interpretation of the minimal GRN as the computational core driving the dynamics of the entire GRN.

In both cases, there is a *master SCC* regulating the other SCCs. The general flow of information can be described like this: external signals enter the SCCs, through the set of controller genes (regulatory genes participating in one-, two-, or three-regulations of SCCs), where they are fed to the logic circuits, logic computations occur within the SCCs, and the signals then emanate outward from the SCCs and from the *master* SCC to the other SCCs. The output of the minimal network is then propagated outward to the fibers (clusters of co-expressed genes) regulated by the SCCs in the periphery of the network (see Fig F in S1 Text) and to other parts of the cellular network, such as the metabolism. Thus, the SCCs act as decision-making units that activate the fibers under their control.

Together, the method lead to the identification of the function for every single gene in the minimal GRN as belonging to three general classes of genes:

- (1) A set of synchronized symmetric fibers
- (2) Regulators of the SCCs

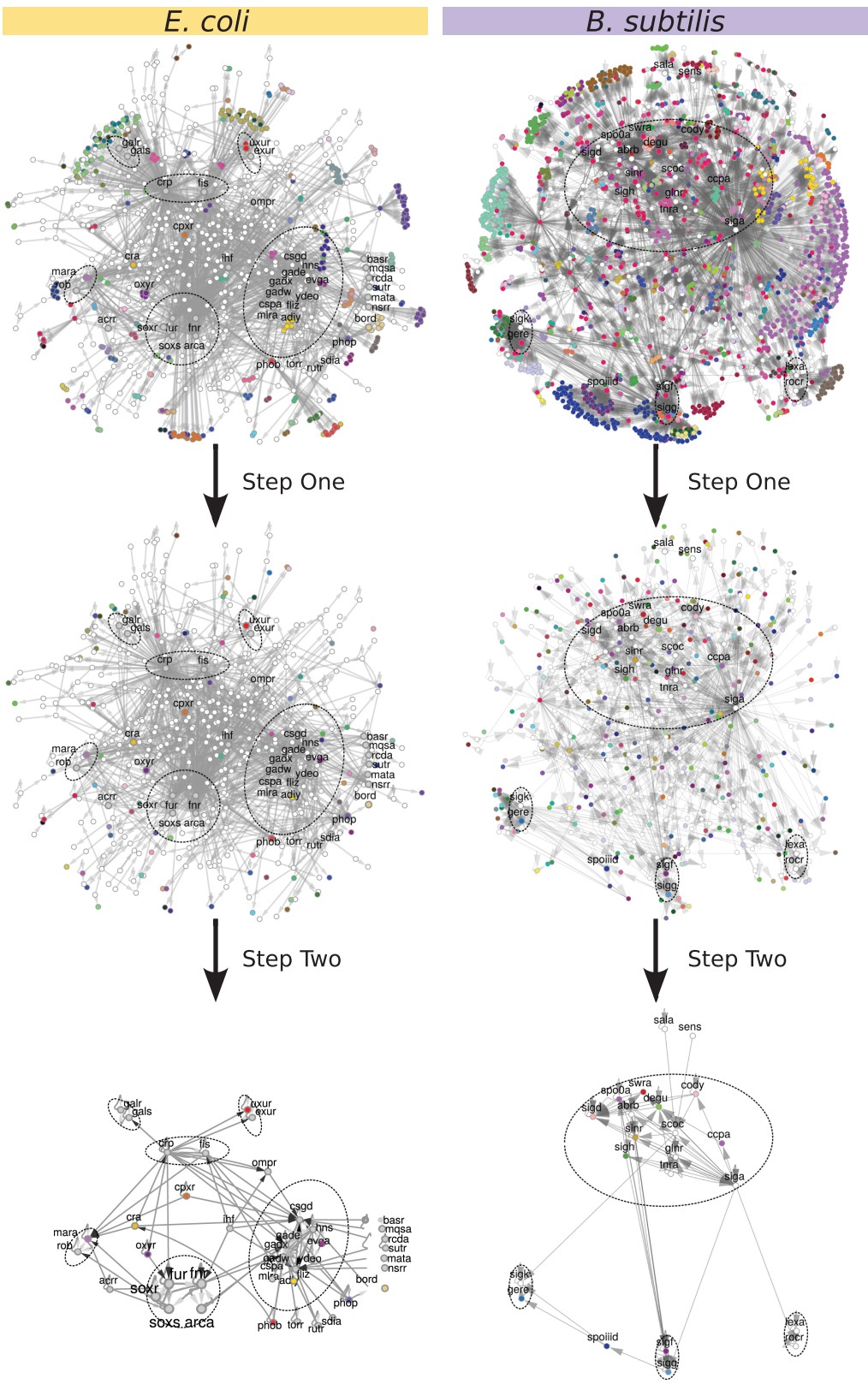

**Fig 4. Network reduction of the GRN of _E. coli_ and _B. subtilis_.** For _E. coli_'s network, we start with the network from Fig 1, rearranged to show the outward flow of signals from the minimal network and with genes in the same fibers colored the same. SCCs are enclosed by ellipses. Genes names are shown only for genes that are part of the minimal

network. Most of the genes belonging to fibers can be seen located in the periphery (outer regions) of the network. *Step one* of the CoReSym procedure collapses all fibers into one representative node, resulting in the base network obtained from the minimal surjective fibration. *Step two* uses a *k*-core decomposition to removes all the dead-end paths ending at nodes with no output, resulting in the minimal network, with only 42 nodes for *E. coli* and 22 in *B. subtilis*. Both minimal networks have a master SCC that regulates the rest, connector nodes connecting different SCCs as well as controller nodes sending inputs to the SCCs.

**Table 1. Gene counts of original and reduced GRNs.** The full genomes for *E. coli* and *B. subtilis* contain 4,690 genes (according to RegulonDB [12]) and 6121 genes(obtained from SubtiWiki [45]), respectively. Among all genes only 1843 and 2482, respectively, express TFs with known interactions. The first reduction step in *E. coli* was performed by a trivial fibration (collapsing the operons, which are trivial fibers), before applying CoReSym. This could also be seen as a part of Step 1 and as such is not needed for *B. subtilis*.

| | *E. coli* | | *B. subtilis* | |
|---|---|---|---|---|
| **Reduction step** | *Genes* | *%* | *Genes* | *%* |
| Step 0.0: Full Genome | 4,690 | – | 6121 | – |
| Step 0.1: GRN (non isolated TFs) | 1,843 | 100% | 2482 | 100% |
| Step 0.2: operon-collapsed GRN | 879 | 48% | – | – |
| Step 1: Base-GRN (collapsing fibers) | 555 | 30% | 521 | 21% |
| Step 2: Minimal GRN (after pruning) | 42 | 2% | 22 | 0.9% |

**Table 2. Statistics and fiber coverage of the two GRNs.** For *E. coli* we start with the 879 operon-GRN from Step 0.2 (see previous Table 1). For *E. coli*, Step 1 collapses the 416 nodes within fibers into 92 fiber-collapsed nodes (one for each fiber), to give the collapsed-fibers 555 nodes Base-GRN. For *B. subtilis*, the 2263 fibered nodes are collapsed into 302 fiber-collapsed nodes (one for each fiber), resulting in the 521 nodes Base-GRN. Step 2 removes all the nodes in the outer shell, including 82 of the fiber-collapsed nodes for *E. coli* and 290 for *B. subtilis*, thus leaving only the minimal GRN. The minimal networks are composed of the nodes in SCC and the connectors nodes. The *k*-shells are actually much bigger, but we count the genes inside them after collapsing the fibers, resulting in only 82 fiber-collapsed nodes (in the case of *E. coli*) instead of counting all the original nodes that belong in these fibers.

| | *E. coli* | | *B. subtilis* | |
|---|---|---|---|---|
| **GRN breakdown** | *Genes* | *%* | *Genes* | *%* |
| GRN | 879 | 100% | 2482 | 100% |
| Nodes in fibers | 416 | 47.3% | 2263 | 91.2% |
| $k_{out}$ shell (fiber-collapsed nodes) | 513(82) | 58.4% | 499(290) | 20.1% |
| Nodes in SCCs (fiber-collapsed nodes) | 24(4) | 2.73% | 19(11) | 0.77% |
| Connectors (fiber-collapsed nodes) | 18(6) | 2.05% | 3(1) | 0.12% |

**Table 3. List of gene circuits in *E. coli* and *B. subtilis*.** The circuits found in *E. coli* are described in detail in Fig 5 and discussed at length in Section C in S1 Text.

| Circuit type | *E. coli* | *B. subtilis* |
|---|---|---|
| **Toggle-switch type** | *galS-galR, uxuR-exuR, csgD-fliz* | *lexa-rocr, glnr-tnra* |
| **Oscillator type** | *rob ↦ marA, soxS ↦ fur, cspA ↦ hns, gadX ↦ hns,* | *sigk ↦ gere, siga ↦ spo0a* |
| **Lock-on types** | | *sigf-sigg, siga-sigh, siga-sigd, sigd-swra* |
| **Capable of various types** | *crp-fis, gadW-gadX, fnr-arcA* | |
| **FFF type** | *ihf ↦ {fis,fliz} ↦ gadX-gadE ihf ↦ {fnr,fliz} ↦ gadX-gadE ihf ↦ {fnr,fliz} ↦ gadW-gadE* | |

- (3) logic circuits within the SCCs (often arising from an $n = 2$ fiber symmetry breaking), which can be further classified into:
  - (3.1) memory devices (toggle switches)
  - (3.2) oscillators.

Perhaps it is import to note that this result does not depend on the simplification assumptions or even the specific ODEs used to model the genetic interactions, this is purely the network structure. The specific details of the parameters affect how large the synchronization within the fibers is, while the choice of function used determines the specific activation levels, times, and bifurcation processes that impact the specific details of the internal computations of the logic circuits; however, the overall flow and decomposition of the network is a result of network structure alone.

The code implementing the method for reproducing the entire analysis in the present paper can be found in https://github.com/luisalvarez96/MinimalTRN and a more detailed explanation as well as a pseudocode can be find in Section F in S1 Text.

## 3.2 Minimal gene regulatory network of *Escherichia coli*

The gene regulatory network of *Escherichia coli*, as provided by RegulonDB, contains around 4,690 genes; however, the majority of these genes code for proteins that do not regulate any other genes, but are enzymes, structural proteins, etc., or they may also constitute TFs whose interactions are not yet well known. Since these genes do not have a clear regulatory role within the GRN, we do not consider them in our analysis, and keep only the annotated transcription factors. This leaves us with 1,843 genes involved in the GRN.

**3.2.1 Network reconstruction and reduction to transcriptional units (TUs).** Operons are clusters of contiguous genes that get transcribed together by the same polymerase as a unit, hence being "trivially" synchronized. Some operons however, have internal promoter regions that allow some of the genes of the operon to be transcribed in different transcription units (TUs) without the need to transcribe the full set of genes in the operon. To simplify our analysis, we also collapse transcriptional units (genes in the same operon, under the control of the same promoters) operons into single nodes, because such genes would form "trivial" fibers. To do so, operons with such internal promoters are split into its different transcription units (TUs) in our network, which are now treated as gene nodes. This trivial reduction, which is in reality a part of our step 1 of collapsing, shrinks our initial 1843-genes GRN to 879 nodes. Strictly speaking, this reduction is part of our Step 1 and can be done together in only one step, however, we took this step to avoid looking at operon's trivial fibers, during initial analysis (as was done in Ref. [10]).

**3.2.2 Network reduction.** Applying the symmetry fibration to *Escherichia coli*'s GRN results in just 555 nodes, 30% the size of the original network, as seen in Table 1. Step 1 could also have been applied directly to reduce the initial 1843 network to the 555-node base network. Most of the genes belonging to fibers are located in the periphery of the network. Various nodes with notably high out-degree in the network don't belong to the core network. After this, the remaining 555 nodes are reduced by taking the $k_{out} = 1$-core of the network into just only 42 nodes. These 42 nodes make up the minimal GRN which corresponds to the computational core of the GRN.

**3.2.3 Large-scale structure of *E. coli*'s minimal GRN.** The structure of the core network (as shown in Fig 1 on the right), is obtained from the reduction process illustrated in Fig 4. We find that *E. coli*'s minimal GRN is composed of 6 SCCs, see Fig 4.

The central subunit of this minimal GRN is the carbon SCC: *crp-fis*. Which serves as the carbon utilization subnetwork [12] controlling a set of TF and enzymes involved in the catabolism of the different sugars and thus is one of the main components for the life of the cell. Another SCC, with 5 nodes, involved in responses to *stress* [12] in the cell. We call this the *soxS* SCC, made up of *fur-fnr-arca-soxs-soxr*, bottom center of the minimal GRN in Figs 4 and 5. Additinally, we obtain one with 11 nodes which mostly regulates the cell's *pH* response [12] (we will refer to it as the *pH* SCC, lower right corner of the minimal GRN in Figs 4 and 5) The remaining three SCC are: *marA-rob* SCC, which controls a number of genes involved in resistance to antibiotics [12]; the *uxuR-exuR* SCC, which are involved in regulation of the transport and catabolism of galacturonate and glucuronate [12]; and the *galR-galS* SCC related to the import and catabolism of galactose [12]. For a detailed description of the signal vortices, see Section D in S1 Text.

The three main SCCs are connected in a forward manner: carbon → ph, carbon → stress, and stress → ph; thus forming a feedforward loop representing the core of the genetic computing system. The same structure is observerd between the carbon SCC, the stress SCC and *marA-rob* SCC (carbon → *marA-rob*, carbon → stress; stress → *marA-rob*). The two remaining SCCs only receive information from the carbon SCC.

**3.2.4 Logic circuits present in *E. coli*'s minimal GRN.**  Inside the SCCs of the GRN we found various genetic circuits that resemble logic circuits designed and implemented in the synthetic biology literature. In total, 12 different pairs of genes were found to be involved in a number of logic circuits. Fig 5 shows all the circuits found in *E. coli*'s GRN, as well as all the inputs to them that break their symmetry and drive their computing. For a detailed description of the gene circuits and what the literature tells us about their possible dynamics, see Section D in S1 Text.

**3.2.5 Simple directed cycles in *E. coli*'s minimal network.**  In total we found 41 simple directed cycles in the minimal core of *E. coli*'s GRN. Four of them are the two-node SCCs, 5 are located in the *soxS* SCC and the remaining 32 are located in the *ph* SCC. Each of the cycles contains at least a pair of nodes that between them form a logic circuit. This means that each cycle longer than 2 nodes passes through at least two nodes that are also connected by a logic circuit (all logic circuits themselves are, of course, two-node cycles). For example, in the case of *soxS* SCC we observe 2 two-node cycles (circuits *soxS* ↦ *fur* and *fnr-arcA*, see Fig 5C), while the remaining 3 cycles in this component all pass through *soxS* ↦ *fur* and as such can be considered longer loops from *soxS* to *fur*, this is shown in Fig 6 on the left. In many case, the cycles even pass through multiple nodes that are connected by a logic circuit to each other. This is also visible on the right of Fig 6, where the longer loops passthrough *gadE-gadW*, *gadW-gadX*, and *gadX-hns* all of which are different logic circuits by themselves. All of the loops illustrated can be considered loops of various lengths between the circuits *soxS* ↦ *fur* and *csgD–fliz*. A complete list of all cycles is available in the provided repositories.

## 3.3 Minimal gene regulatory network of *Bacillus subtilis*

Next, we analysed the GRN of a second example organism, the soil bacterium *Bacillus subtilis*. *B. subtilis* is a main model organism for Gram-positive bacteria with a well-studied GRN. The data used for *B. subtilis* included not only transcriptional activators and repressors, but also sigma factors, which play a bigger role in this organism [43]. A sigma factor binds to the promoter region of a target gene to enable its transcription, and different sigma factors

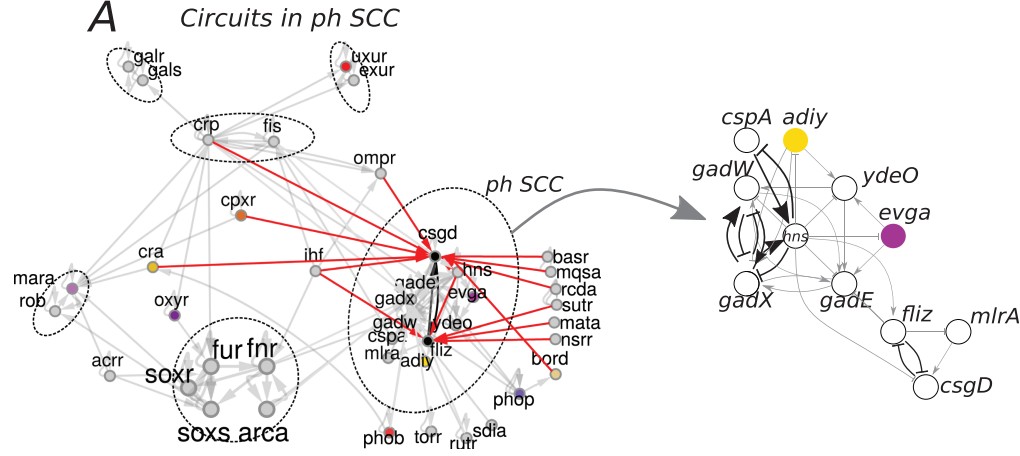

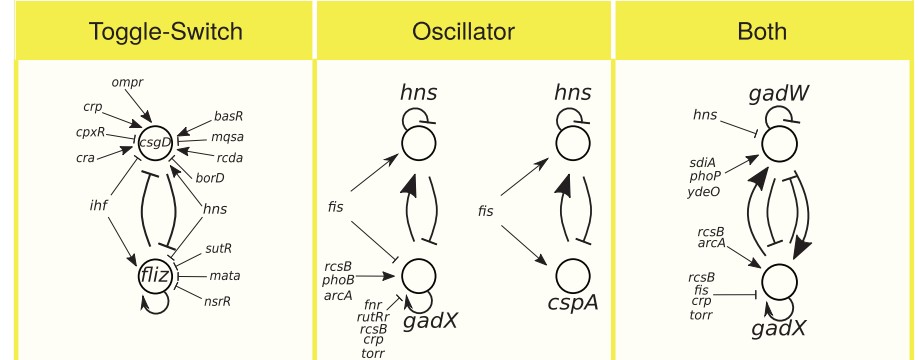

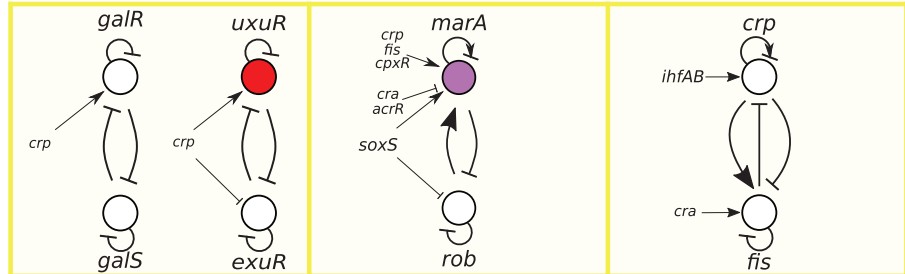

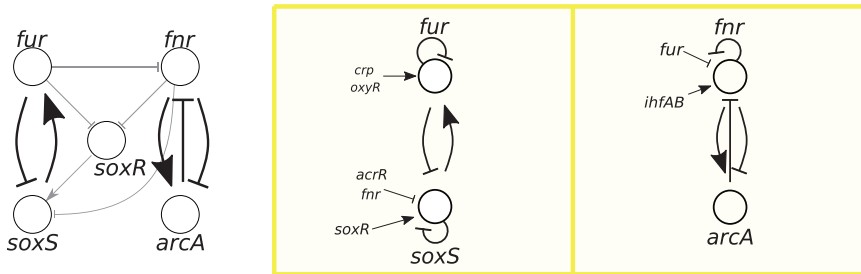

**Fig 5. Circuits in the GRN of *E. coli*.** (**A**) The minimal GRN of *E. coli* and the circuits embedded in it, shown with red links for the symmetry breaking inputs to the toggle-switch *fliz-csgd*. The biggest SCC is in charge mostly of pH

responses. Colored nodes represent fibers. (**B**) The two-node SCCs and (**C**) the *soxS* SCC and its circuits. For each circuit, the incoming signals that break the symmetry are shown.

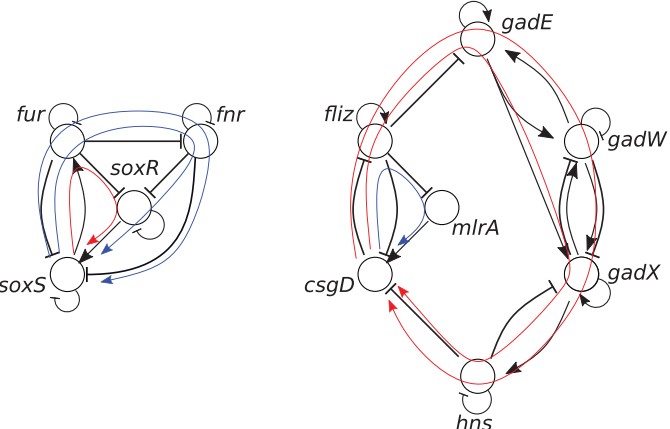

**Fig 6. Some simple directed cycles in *E. coli*.** The networks shown are different cycles that cross through the logic circuits *soxS-fur* (left) and *csgD-fliz* (right). Arrow colors denote the overall sign (overall activation: blue; overall inhibition: red).

target specific groups of genes, for example, genes involved in stress responses. In our analysis, we treated sigma factors as inducers, just like activating specific TFs. Previously, the integrated metabolic and regulatory network of *B. subtilis* has been broken down into functional modules, locally regulated or regulated by a global regulator [43,44]. However, the overall structure of this network, how the modules interact with each other, and the overall flow of information or signals between these are still not entirely understood. Similarly as in *E. coli*, our method revealed a structure of the gene regulatory network for this bacteria and to identify the possible logic circuits at the core of the network. Like for *E. coli*, all nodes in the GRN could be classified: in the original GRN, each node either belongs to a fiber, belongs to a logic circuit, or sends inputs to circuits.

**3.3.1 Network reduction.** The first reduction by fibration reduces the network size to 21% of its original size. The *k*-core reduction led to a further reduction to just 0.9% of the original nodes, as can be seen in Table 1. The resulting core GRN for *B. subtilis* is shown in the right column of Fig 4 and corresponds to just 22 nodes.

**3.3.2 Large-scale structure of *B. subtilis*'s minimal GRN.** The structure of the minimal network is shown in Fig 7. Like in *E. coli*, the resulting minimal set of TFs obtained for *B. subtilis* contains only the SCCs, the fibers that connect them and the nodes that send signals to control them. However, it is particularly interesting how this minimal gene regulatory network is smaller, with just 22 nodes in 4 SCCs, than the one for *E. coli*, given that the original network is larger (2482 nodes). In contrast to *E. coli*, *B. subtilis*' minimal GRN is almost exclusively the SCCs with only 3 controlling nodes whereas in *E. coli* there are 18 controlling nodes that inform the SCCs modules.

The structure of the minimal computational core in *B. subtilis* is simpler than in *E. coli*. It consists of only 4 SCCs: the *siga* SCC, a large central SCC as a hub composed of 13 nodes, which regulates the other three SCCs, each containing only two nodes. This SCC hub is controlled by two control nodes (*sala* and *sens*). Importantly, we also find a feedforward

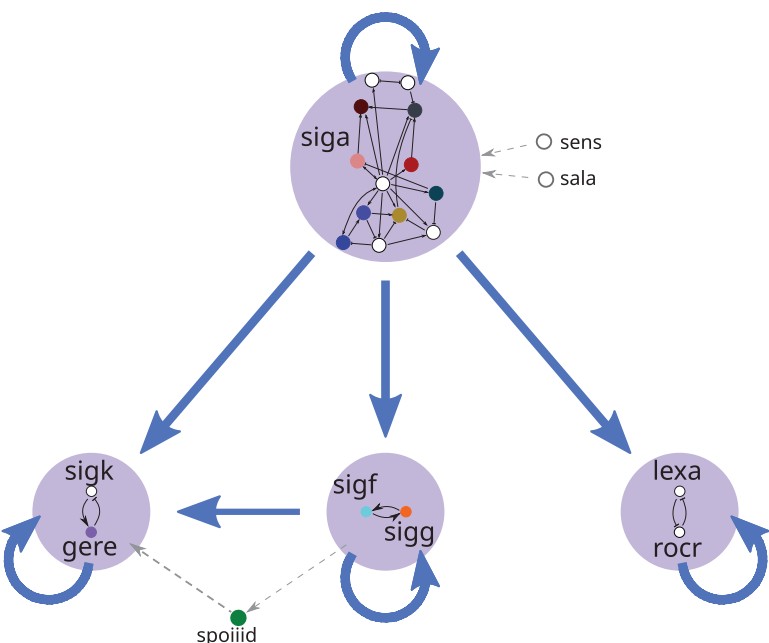

**Fig 7. Sketch of reduced GRN of *B. subtilis*.** The 4 SCCs are shown: *siga* SCC, *sigk-gere* SCC, *sigf-sigg* SCC, and *lexa-rocr* SCC. The signaling flows between them: with the *siga* SCC the hub controlling the other three SCC and being fed information signals by the two controllers *sens* and *sala*, controller node *spoiiid* connects the *sigk-gere* SCC and *sigf-sigg*. A feedforward structure between the *siga* SCC and the *sigk-gere* SCC is visible.

structure with the master regulator SCC regulating both the *sigf-sigg* and *sigk-gere* SCCs. This second SCC also receives signals from *sigf-sigg*. The SCC *lexa-rocr* receives only from the central SCC, and this functions as the input to the circuit *lexa-rocr*. Two controller nodes feed directly into the central hub, while the third controller node connects the *sigf-sigg* to the *sigk-gere* SCCs.

**3.3.3 Logic circuits present in *B. subtilis*'s minimal GRN.** We found only 8 logic circuits in *B. subtilis*, less than the 12 found in *E. coli*. Two of them correspond to MR circuits (toggle-switch type): the *lexa-rocr* SCC and *glnr-tnra* from the central SCC. Like in *E. coli*, both of these circuits present additional negative autoregulations in each gene (self-inhibitions, see Section D in S1 Text for a discussion on the effect of these self-loops). Two other circuits are NFBL circuits (oscillating types): the *sigk* ↦ *gere* SCC, which actually corresponds to an amplified NFBL given that *sigk* possess a self activation and *siga* ↦ *spo0a* from the central SCC, actually corresponds to a Smolen oscillator because of its self-loops. The main difference with respect to *E. coli* is that we observe 4 positive autoregulation (PAR) feedback loop circuits, possibly *lock-on* circuits: the *sigf-sigg* SCC and the rest within the central SCC. All of them contain at least one gene with an additional positive autoregulation. In *E. coli* the only *lock-on* feedback loops present were part of the *feedforward fiber* type circuits. See Table 3 for more details.

**3.3.4 Simple directed cycles in *B. subtilis*'s minimal network.** The number of cycles in *B. subtilis* is a bit larger than in *E. coli*. Out of the 48 cycles, 3 correspond to the two-node SCCs and the remaining 45 cycles are located within the central SCC. As in *E. coli*, all the cycles, except for one, pass through nodes also connected by logic circuits. The one exception

is the cycle formed by *abrb-sigh-spo0a-abrb*, although *sigh* and *spo0a* do belong to logic circuits, they are not part of the same circuits. Again, many cycles pass through multiple nodes that are connected by a logic circuit, as shown in Fig 6 for *E. coli*.

### 3.4 Statistical significance of the network structures observed

If we find structures in graphs, we may ask whether these structures are expected in graphs of a certain type, or whether their observed numbers are unexpectedly high. In biological networks, this reflects a similar question: are the structures observed expected to appear in evolution just by chance, or are they so "unlikely" that we need to assume that they were favored by evolution for some functional benefit? To see whether the observed structures do not only emerge by chance, but are functionally relevant, we compared the *E. coli* and *B. subtilis* networks to corresponding "null hypothesis" ensembles of random graphs, following the recipe used to find significant network motifs. Our random network are supposed to represent the hypothetical outcomes of an evolution based on mutations, but without a selection for function; edges are rewired, while preserving some basic structure of the original networks (in particular, the in- and out-degrees of all individual nodes). Structures that appear in the real networks, but are rare or absent in randomized networks can be assumed to be due to an evolutionary selection, probably for specific functional advantages. To assess the statistical significance of the structures observed, we followed a standard approach: we compared our results to results from randomized networks, representing a null hypothesis. Details and some quantitative results of the analysis are given in Section E in S1 Text.

The results are shown in Fig 8. Almost all the studied structures found in both bacteria are statistically significant, as shown by their Z-score on Fig 8B (Table A in S1 Text provides a full breakdown). All fiber classes are significantly over-abundant in the real networks, with the exception of the simplest fiber building block of only one regulator, $|n = 0, \ell = 1\rangle$, which is significantly absent. This suggests that evolved GRNs favor more complex wiring patterns, more complex fiber building blocks than just trivial ones, that allow for richer dynamics and more flexible control.

Our analysis suggests that the two bacterial networks resulted from an evolutionary selection for specific functional structures. This concerns both the large-scale substructuring of the core into several SCCs and the variety of small-scale circuits, which is far richer than that to be expected with only random mutations at play.

### 3.5 The bacterial minimal GRN as a computation device

Having a clear picture of their structure, we can now describe bacterial gene regulatory networks as a computation device. The two primary components of computer processors are flip-flops (toggle-switches) and oscillators, both of which are also present in our minimal bacterial GRNs. Thus, the GRNs studied can be seen as computational devices in which data are stored and processed by broken symmetry circuits within the SCCs. The flip-flops in the SCCs control how fibers are turned on and off, and the symmetric fibers themselves represent clusters of genes displaying coherent synchronization in gene co-expression. The results of these computations in the flip-flops is then transmitted to other parts of the cellular network through the fibers. The switching of the flip-flops between their stable states (i.e., zero and one) itself is controlled by a set of controller genes that regulate the SCCs (red arrows in Fig 5A for example). These controller genes can regulate one, two, or three SCCs simultaneously. Hence, decision making emanates from the SCCs who then turn on and off the fibers under their control. So, the SCCs are crucial as the decision-making units of the GRN.

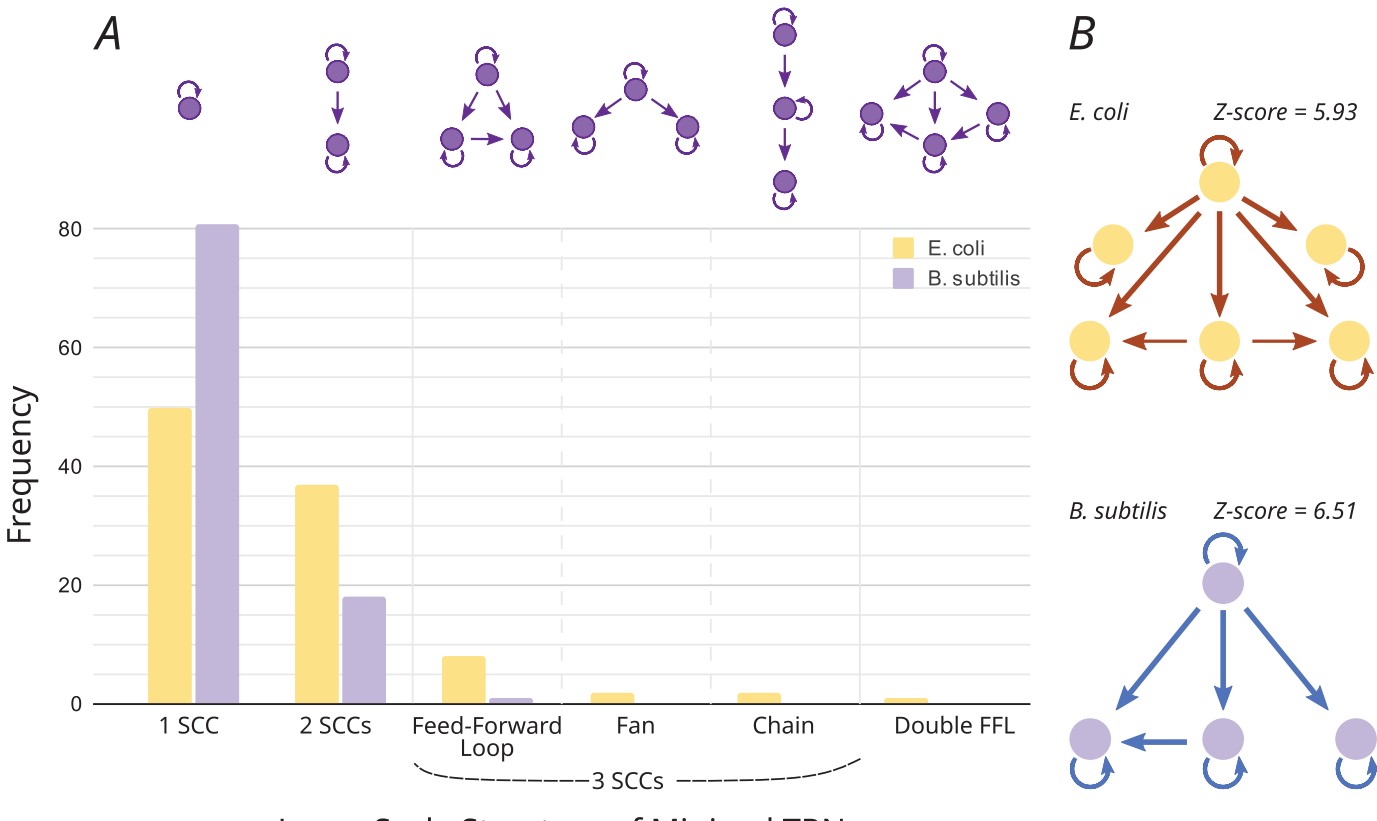

**Fig 8. Statistically significant large-scale structures in *E. coli* and *B. subtilis* core GRNs.** The structure between SCCs is compared to the corresponding structure in randomized networks with the same in- and out-degrees of all nodes (and preserving the edge types). **A)** Histogram illustrating the distribution of the observed structures in the core of the random networks along with a sketch of the structure itself atop the histogram. Following the format of Fig 1, purple circles represent SCCs and arrows stand for edges between SCCs. **B)** Structures observed in *E. coli*'s and *B. subtilis*'s core shown with the *Z-scores* of obtaining a structure with such number of SCCs from the randomized networks.

Since all logic genetic circuits are contained in our SCCs, our reduction method preserves the entire computationally relevant circuitry of the original network. The controlling nodes to the SCCs act as symmetry breaking nodes giving rise to the symmetry broken circuits, more generally, as inputs for the logic genetic circuits within the SCCs. We would anticipate that, unlike carefully designed and implemented circuits in synthetic biology, external inputs to the feedback circuit structure are crucial to the computational and biological behavior of observed circuits and to the entire *computational apparatus* network and the bacteria itself. This is why it is important to understand the topology of the whole network and the communication between the different modules of the network among themselves and with the extra-cellular environment.

Hence, overall, we can now describe the dynamics of the network in the following way: the computational core, i.e., the resulting minimal GRN, "executes" a response to its input signals coming from metabolism and from the extracellular environment; its outputs are propagated through the network in a signaling cascade-like event, through the fibration tree-like structure, flowing outward of the computational core to the peripheries of the network.

## 4 Discussion

### 4.1 Network structures revealed by CoReSym

In gene regulatory networks, signaling is decentralized: while there are some "mighty" master regulators, there is no single central agent that ultimately controls the expression levels of all genes. However, we identified here a computational core of the network consisting of a number of "vortices" in which signals can cycle and which are connected to each other in a feedforward fashion. This core network, or minimal GRN, shows an interesting modular structure, the large-scale structure of the minimal GRN. We interpret this subnetwork as the *core computational apparatus* of the network, composed of an ensemble of logic genetic circuits. Furthermore, these vortices are internally composed of various logic circuits on the smaller scale. These logic circuits perform computations that drive the dynamics, while signals then propagate through the fibers to the rest of the network. The nodes in the peripheries receive signals from the core, modify their shapes, and relay them to output nodes.

Previous works on bacterial GRNs have either focused on local motifs and larger modules arising from their integration (i.e. FFL and Dense Overlapping Regulons) or on the co-regulation of functionally related genes. Both views ignore the arrangement of genes in the overall network structure. In contrast, our fibration analysis considers the entire network and identifies the most influential circuits on the basis of their placement at the core of the network. In the bacterial network studied, we find a rich interaction between the network's SCCs as well as the important role played by feedback loops both in determining the SCCs and the circuits within, compared to previous observations [18]. Following our method, all genes in the GRN can be classified with regards to the function they perform computationally in the message passing dynamics. Each gene in the GRN belongs either to a fiber or to some logic circuit or sends inputs to logic circuits. Most of the external regulators of SCCs are directed towards logic circuits serving as the inputs for their computations.

Our study into the GRNs of the model bacteria *E. coli* and *B. subtilis*, after network reduction, highlight the structures that control their dynamics, and show how signals can propagate in the network. "Computations" occur only in a subset of nodes, while the others, towards the periphery of the network, just aggregate and modulate output signals. On a large scale, the core network consists of a number of "signal vortices" in which information can cycle and which are connected by feedforward arrows. On the small scale, we found logic circuits which again may have either a feedforward shape, transmitting signals only in one direction, or contain feedbacks (allowing for permanent inner states).

Our fibers account for global signal flows in the network that reach the symmetric nodes within a fiber, which allows us to compare nodes by the global signals they receive. While the core network comprises the minimal set of nodes driving the rest of the network, the logic circuits, embedded in the core's SCCs, are located in influential positions at the core of the network. Importantly, we not only take into account the feedback loops in these networks, a key feature that has often been missed in previous analysis, but we show them to be crucial to the decision-making and computational abilities of the network since they are what defines the logic circuits and the modules in our breakdown.

### 4.2 Biological relevance of the structures found

Since any network can be dissected into such structures, this raises the question whether the structures we observed are biologically meaningful or arise just by chance. A first way to test this is to check whether they are statistically significant. In fact, the numbers and sizes of vortices found in the GRNs were significantly different from the numbers and sizes expected in

random graphs. Applying the same analysis to randomized networks with the same degree distribution as both GRNs studied, we found that these randomized networks tend to show a much simpler overall structure, consisting of only one (much bigger) vortex. In contrast to this, we found that the number of SCCs has a Z-score of 5.93 in *E. coli*, and of 6.51 in *B. subtilis*. Along with the number of circuits observed, also having high Z-score values (see Section E in S1 Text for more details.) The observed structuring of bacterial networks into separate vortices therefore seems to be a result of selection advantages in evolution, which suggests a biological function.

In the past, modularity and community detection, e.g. the Louvain algorithm [46], have been used to partition biological networks into functionally coherent modules. In our CoReSym scheme, instead, we partition the network differently. By using a combination of fibers and SCCs we are able to break down the network to its minimal structure. The SCCs comprise structures that are functionally coherent, i.e., each SCC is mostly associated with a single type of cellular function, from sugar consumption to stress. Co-occurrence of genes in an SCC seems to be a better predictor of shared biological function than co-occurrence in a module determined by community detection algorithms. Furthermore, SCCs control a set of fibers that – assuming equal gene regulatory input functions – comprise co-expressed sets of genes, which may correspond to shared biological function.

All these (statistically significant) structures are revealed by our method and would not be visible otherwise: once the network was simplified by our graph fibration, the SCCs emerge almost by themselves, and the network structure looks suddenly simple and comprehensible! This may be a lesson for understanding other, maybe even less structured and more dynamic networks modeling other forms of collective intelligence.

## 4.3 Gene logic circuits: gene duplication and symmetry breaking

The emergence of network motifs [13–15] has been explained by an evolutionary setting of mutation and selection, with a random rewiring of the GRN (mutation) and a selection for functional structures. While this mechanism could also explain the appearance of gene fibers, it would probably be rather slow. Another much faster genetic mechanism, gene duplication, may be able to generate fiber structures in evolution fast and almost "for free." Minor modifications in the duplicated nodes could then happen by subsequent mutations.

Gene duplication is an important evolutionary process that results in a cell having two (or more) paralogue copies of a gene (or set of genes), i.e. a set of *duplicate* genes. When this process results in a set of (duplicated) genes that share the same input relations, it "creates" new fibers organically. The duplicated set of genes belongs to a synchronous fiber. In this case, the duplication itself works as the lifting property, where the fiber node in the base is "lifted" to the nodes that belong to the fiber. In this way, gene duplication offers a plausible explanation as to why the systems studied here exhibit so many symmetries, in that so many of its nodes belong to fibers.

Not only can this explain the existence of so many nodes in fibers, but furthermore the symmetry breaking of the resulting duplicated building blocks (such as the building blocks depicted in Fig 2) leads to logic gene circuits. For example, by duplicating a self-inhibiting gene in such a way that its input tree is preserved, we obtain two mutually repressive genes, which correspond exactly to the *toggle-switch* flip-flop [20]. Further mutations to each gene that could add additional regulators, separately for each gene, would correspond to the *set-reset* (*S-R*) switches of the toggle switch. This is explained in more detail in Section C in S1 Text.

## 4.4 Limitations of the CoReSym procedure

What are the limitations of our network reduction method? First, bacterial signal processing does not depend on network structure alone, but on quantitative gene regulatory input functions, with parameters that differ between genes in a fiber. Even in a simple threshold model, two genes that receive inputs from the same transcription factors would show different outputs because of different logic operations (e.g., AND versus OR) or of different activation or repression thresholds (whereby one gene may be activated faster than others). As discussed in [11], our fibration analysis does not address this complexity: Instead, it assumes that all genes in a fiber share exactly the same regulation; differences in gene regulatory functions, post-translational regulation, as well as mRNA and protein degradation are not taken into account.

How can we justify this? In the spirit of physics, a symmetrized system can be seen as a first-order approximation, revealing some important general features, in this case, of gene regulation. In this view, considering individual gene properties would be a second step in which we introduce a weak symmetry breaking or second-order approximation, which changes the predicted behavior and makes it more realistic. Although the second step is important to approach biological reality, the first step may provide important insights, while fully detailed, dynamic models of GRNs would prevent us from seeing the forest for the trees. Moreover, the successful implementation of genetic circuits in synthetic biology, capable of basic logic computations [20] such as memory storage and time-keeping by oscillations, suggests that analogies to logic circuits may help us understand decision making also in wild-type cells.

A second limitation of our method comes from the fact that GRNs are linked to the rest of the cell. Other forms of regulation including signaling, small-molecule regulation, and metabolic pathways also display symmetries. In theory, to trace cellular signaling flows and decision processes, a fibration analysis should include not only transcriptional regulation, but also metabolism. In our analysis, metabolites are seen as given inputs to the GRN that may modulate TF activities. In reality, metabolite concentrations depend on enzyme activities, which themselves depend on the GRN, so metabolic and regulatory networks form a large feedback loop. In a fibration analysis of the entire cell, all these networks would need to be combined. The outward pathways, which are eliminated by our *'pruning'* step, may then feed back on the GRN through metabolites that can bind transcription factors as effector molecules and modulate their activities. This could be described by assuming that these metabolites can turn on and off the edges in the GRN. Enzyme phosphorylation, which involves binding a phosphate group to activate or inactivate an enzyme, is another regulation mechanism that acts between GRN and metabolism and could be considered in a larger regulatory network.

**4.4.1 Future challenges.** Despite some progress in this direction [47], some challenges remain. Since metabolic reactions can have multiple substrates and products, metabolic networks have to be treated as hypergraphs [48,49]. Fibrations of hypergraphs still need to be developed. Another challenge concerns the different time scales. Metabolic dynamics is much faster than gene expression dynamics, so on the time scale of gene regulation, metabolism is close to a steady state. In this quasi-steady state, metabolite levels effectively depend on enzyme activities via long-range, non-sparse interactions [50], which complicates a fibration analysis.

Finally, the ideas presented here may inspire the design of artificial GRNs with functionalities of living cells. One could start with the design of the core computational apparatus of the GRN, integrating the desired amount of logic genetic circuits, and then, based on this core network, construct "signal highways" to peripheral genes whose products perform the

required biological functions. In the same vein, CoReSym may also facilitate the design of minimal genomes [51].

## ORCID iD

Hernán A. Makse: https://orcid.org/0000-0001-6474-1324
Wolfram Liebermeister: https://orcid.org/0000-0002-2568-2381
Luis Alvarez-Garcia: https://orcid.org/0009-0008-0229-2461

## Supporting information

**S1 text.** Supplemental information and formal definitions; detailed methodology breakdown; supporting results; algorithms explanation and pseudocode.
(PDF)

## Author contributions

**Conceptualization:** Luis A. Álvarez-García, Ian Leifer, Hernán A. Makse.

**Data curation:** Ian Leifer.

**Formal analysis:** Luis A. Álvarez-García.

**Funding acquisition:** Hernán A. Makse.

**Investigation:** Luis A. Álvarez-García.

**Methodology:** Luis A. Álvarez-García, Wolfram Liebermeister, Ian Leifer, Hernán A. Makse.

**Project administration:** Wolfram Liebermeister, Hernán A. Makse.

**Resources:** Ian Leifer.

**Software:** Luis A. Álvarez-García, Ian Leifer.

**Supervision:** Wolfram Liebermeister, Hernán A. Makse.

**Validation:** Luis A. Álvarez-García, Hernán A. Makse.

**Visualization:** Luis A. Álvarez-García.

**Writing – original draft:** Luis A. Álvarez-García, Wolfram Liebermeister, Hernán A. Makse.

**Writing – review & editing:** Luis A. Álvarez-García, Wolfram Liebermeister.

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
