## [Decision Letter · Decision Letter 0]

7 Aug 2024

Dear Mr Alvarez-Garcia,

Thank you very much for submitting your manuscript "Complexity reduction by symmetry: uncovering the minimal regulatory network for logical computation in bacteria" for consideration at PLOS Computational Biology.

As with all papers reviewed by the journal, your manuscript was reviewed by members of the editorial board and by several independent reviewers. In light of the reviews (below this email), we would like to ask for a significantly-revised version that takes into account the reviewers' comments.

The reviewers raise several critical points to address, including better justifying and explaining assumptions (which may affect the applicability of the findings), making the paper more concise, and clarifying inconsistencies in the text. The authors are encouraged to clearly address all points raised.

We cannot make any decision about publication until we have seen the revised manuscript and your response to the reviewers' comments. Your revised manuscript is also likely to be sent to reviewers for further evaluation.

Sincerely,

William Cannon

Academic Editor

PLOS Computational Biology

Stacey Finley

Section Editor

PLOS Computational Biology

Reviewer's Responses to Questions

**Comments to the Authors:**

Reviewer #1: In this study, the authors develop a method to reduce the complexity of transcriptional regulatory networks based on relaxed principles of symmetry and k-core decomposition. The minimal TRN is decomposed into strongly connected components; symmetry breaking and simple cycles are sought to describe its most basic structures.

It was difficult for me to read and understand the study because it is a large manuscript—potentially excessively large—and readers can benefit from the most concrete writing.

Given my biological background, it is difficult to appreciate the potential utility of the generated knowledge and how this can contribute to advancing the explanation of how these model microorganisms process the environment's signal and are processed internally to respond adequately.

Some sections, like antecedents, methods, and results, are mixed. Some suggestions to improve the manuscript are:

Lines 1409-1411 should be part of methods.

Lines 1481-1483 should be part of the methods. Moreover, add a description of how the random networks were created.

When analyzing TRNs in E. coli and B. subtilis, it is suggested that the subtitles be more descriptive for each case; for example, there are two subtitles called "Cycles."

Examples of applying concepts to solve problems in other areas can be irrelevant to the present work. They may contribute to a shorter manuscript, be avoided, or only mentioned succinctly. For instance, lines 29-32 and 21-25

Some figures, for example, 8, indicate the type of regulation (positive, negative, or dual), but the methodology does not describe whether the type of regulation is considered and, if so, how it is processed.

ComSym's algorithm development seems to be a compilation of already-described methods repeated throughout the entire process. Section 2.7 repeats the previous sections. For instance, lines 1003-1007 are the same as 224-234.

Figures are referred to away from the figure inset; for example, Fig. 7 (page 33) is cited first in the introduction (page 7) and then in section 2.3 (page 18).

In lines 1418-1420, what conclusions were obtained from analyzing the B. subtilis network?

What conclusions are obtained when comparing the structures of the minimal TRN in both species?

Section 3.3.1 can be interpreted as a relationship between the network size and the minimum network size. Is there support for that?

In terms of adaptation, what does the following paragraph mean?

Our analysis suggests that the two bacterial networks resulted from an evolutionary selection for specific functional structures. (Line 1538)

In line 1662, can authors hypothesize what led E. coli or B. subtilis to adopt the structure of their TRN?

In the line "since the main function of the bacterium cell is to process sugars." A reference may be required.

Ideally, the assignment of possible functions (memory, oscillator, and toggle-switch) in section 3.2.2 should be linked with the biological function.

The authors should hypothesize the implications of considering E. coli as a computer. What is the biological logic of designing a minimal synthetic network with the principles presented in the work, and how do they work together? Section 3.2.3

Can an output be generated on line 1561 using a TRN and then propagated into a full TRN, observing the same results as the responses from the full TRN?

The biological role of the primary genes involved in minimal TRN is not described.

Reviewer #2: This paper describes a method for simplifying transcription networks from being very complicated, and completely unintuitive, to being vastly simpler and actually understandable. This is an important step to simplification that has potential widespread application. The authors demonstrate their approach using an E. coli network and a B. subtilis network.

This work described in this paper clearly represents an impressive amount of effort and appears to be significant. However, I am not an expert in this subfield of work and so I had a hard time understanding how much improvement this approach yields over existing methods.

I have three major critiques of the work.

First, the paper is extremely long (at somewhere around 20,000 words) and very mathematical, with minimal connection to actual biology. These aspects make the paper very hard to read. In my case, I have a physics background and reasonably strong math skills, but I still struggled to maintain interest in the work after about the first 20 pages. This was less than halfway through the manuscript. I suggest that the authors shorten the manuscript dramatically, moving substantial amounts of content to online supplementary material.

Second, the manuscript does not discuss the broader field of transcription network simplification, which makes it hard to assess how much of an improvement these new methods provide over existing work. It has many references to closely related work, but even those papers are not described here. Most readers are not going to track down each of these references to gain this background material.

Third, the approximations that are made in this work appear to be large enough that I question if the final results adequately represent real biology. For example, symmetries are powerful in physics because they arise from underlying symmetries in nature. However, this does not appear to be the case in biology; instead, the symmetries identified here appear to arise primarily from the approximations that were made to the systems rather than from the properties of the systems themselves. Thus, I question if the symmetries identified here represent anything that’s particularly interesting, or if they are just an applied method for simplifying transcriptional networks. As a related issue, this work appears to build on Boolean approximations for transcriptional networks, which have been known for a long time to be extremely coarse and not particularly accurate.

As a minor issue, this paper has numerous spelling mistakes. I suggest doing a careful proofread and also running it through a spell-checker.

Reviewed by Steve Andrews

Reviewer #3: This manuscript presents an approach for reducing ODE based networks to a minimal version that is referred to as the k-core of the network. The k-core is composed of a collection of strongly connected components. The method is based on collapsing nodes that belong to the same equivalence class using symmetry fibrations and by k-core decomposition. The reduced version of the network preserves the dynamics of the original network. The authors applied their reduction approach to two networks from bacteria (e. coli and b. subtilis).

I find this manuscript mostly well-written, and their approach could be useful to analyze other networks. However, some details of the method are missing and there are some inconsistencies in the name of the method. Therefore, I suggest the following revisions before recommending it for publication.

Major revisions:

1. The method is presented as a list of 5 steps. It’d be better to provide a pseudo-code of the reduction algorithm, so others can implement it.

2. Is the method called “ComSym” or “CoReSym”? In Section 2.7 it is presented as ComSym and later as CoReSym.

Minor revisions:

1. The panel labels (a) and (b) in Fig. 1 are missing.

2. On line 553, page 14, it says that “nodes 1 and 4 have no inputs”, but node 1 has input from node 2, right? It is even shown in Fig. 3B.

3. Reference [38] is a preprint from 2013. Has this been published? If so, please update this reference.

4. Change “fibation” to fibration on line 865.

5. Check formatting of reference [31].

6. In the caption of Fig. 7, add a space before “Step two removes …”.

**Have the authors made all data and (if applicable) computational code underlying the findings in their manuscript fully available?**

Reviewer #1: None

Reviewer #2: None

Reviewer #3: Yes

PLOS authors have the option to publish the peer review history of their article (what does this mean?). If published, this will include your full peer review and any attached files.

Reviewer #1: No

Reviewer #2: **Yes: **Steven S. Andrews

Reviewer #3: No
---

## [Decision Letter · Decision Letter 1]

26 Mar 2025

Dear Mr Alvarez-Garcia,

We are pleased to inform you that your manuscript 'Complexity reduction by symmetry: uncovering the minimal regulatory network for logical computation in bacteria' has been provisionally accepted for publication in PLOS Computational Biology. Please see the comments from Reviewer 1 and revise as needed.

In addition, before your manuscript can be formally accepted you will need to complete some formatting changes, which you will receive in a follow up email. A member of our team will be in touch with a set of requests.

Best regards,

William Cannon

Academic Editor

PLOS Computational Biology

Stacey Finley

Section Editor

PLOS Computational Biology

Reviewer's Responses to Questions

**Comments to the Authors:**

Reviewer #1: The authors have effectively addressed the questions raised and restructured the manuscript to enhance its overall readability and coherence. The discussions on the minimized network structures between the compared species are particularly compelling and serve as a strong illustrative component of the study.

A few minor observations and suggestions for further improvement include:

On line 43, the term "microscopic parameters" could be more accurately replaced with "molecular parameters" to better reflect the context.

On line 57, a line break appears to disrupt the flow of the text and may warrant revision to maintain focus.

On line 522, there is a typographical error in the word "corresponds" that should be corrected.

In Figure 8, it is recommended that the axes be labeled with appropriate titles to improve clarity and interpretability.

These adjustments would further strengthen an already well-presented manuscript.

Reviewer #2: This revised manuscript is much improved and satisfactorily addresses all of my concerns. I recommend publication.

**Have the authors made all data and (if applicable) computational code underlying the findings in their manuscript fully available?**

Reviewer #1: Yes

Reviewer #2: Yes

PLOS authors have the option to publish the peer review history of their article (what does this mean?). If published, this will include your full peer review and any attached files.

Reviewer #1: No

Reviewer #2: **Yes: **Steven S. Andrews

---

## [Editor Report · Acceptance letter]

PCOMPBIOL-D-24-00474R1

Complexity reduction by symmetry: uncovering the minimal regulatory network for logical computation in bacteria

Dear Dr Álvarez-García,

I am pleased to inform you that your manuscript has been formally accepted for publication in PLOS Computational Biology. Your manuscript is now with our production department and you will be notified of the publication date in due course.

With kind regards,

Anita Estes
